# Era3D: High-Resolution Multiview Diffusion Using Efficient Row-wise Attention

Peng Li[1*]   Yuan Liu[2,3*]   Xiaoxiao Long[1,2†]   Feihu Zhang[3]   Cheng Lin[2]   Mengfei Li[1]

Xingqun Qi[1]   Shanghang Zhang[4]   Wenhan Luo[1]   Ping Tan[1,5]   Wenping Wang[2]

Qifeng Liu[1] Yike Guo[1†]
*Equal contribution †Corresponding author

[1]HKUST [2]HKU [3]DreamTech [4]PKU [5]LightIllusion

## Abstract

In this paper, we introduce **Era3D**, a novel multiview diffusion method that generates high-resolution multiview images from a single-view image. Despite significant advancements in multiview generation, existing methods still suffer from camera prior mismatch, inefficacy, and low resolution, resulting in poor-quality multiview images. Specifically, these methods assume that the input images should comply with a predefined camera type, e.g. a perspective camera with a fixed focal length, leading to distorted shapes when the assumption fails. Moreover, the full-image or dense multiview attention they employ leads to a dramatic explosion of computational complexity as image resolution increases, resulting in prohibitively expensive training costs. To bridge the gap between assumption and reality, Era3D first proposes a diffusion-based camera prediction module to estimate the focal length and elevation of the input image, which allows our method to generate images without shape distortions. Furthermore, a simple but efficient attention layer, named row-wise attention, is used to enforce epipolar priors in the multiview diffusion, facilitating efficient cross-view information fusion. Consequently, compared with state-of-the-art methods, Era3D generates high-quality multiview images with up to a $512{\times}512$ resolution while reducing computation complexity of multiview attention by 12x times. Comprehensive experiments demonstrate the superior generation power of Era3D- it can reconstruct high-quality and detailed 3D meshes from diverse single-view input images, significantly outperforming baseline multiview diffusion methods. Project page: `https://penghtyx.github.io/Era3D/`.

## 1 Introduction

3D reconstruction from single-view images is an essential task in computer vision and graphics due to its potential applications in game design, virtual reality, and robotics. Early research [42, 44, 53, 78, 28] mainly relies on direct 3D regression on voxels [39, 58, 9], which often leads to oversmoothed results and has difficulty in generalizing to real-world unseen objects due to limited 3D training data [4]. Recently, diffusion models (DMs) [16, 50] show strong generation ability on image or video synthesis by training on extremely large-scale datasets [47, 48]. These diffusion models are promising tools for single-view 3D reconstruction because it is possible to generate novel-view images from the given image to enable 3D reconstruction.

38th Conference on Neural Information Processing Systems (NeurIPS 2024).

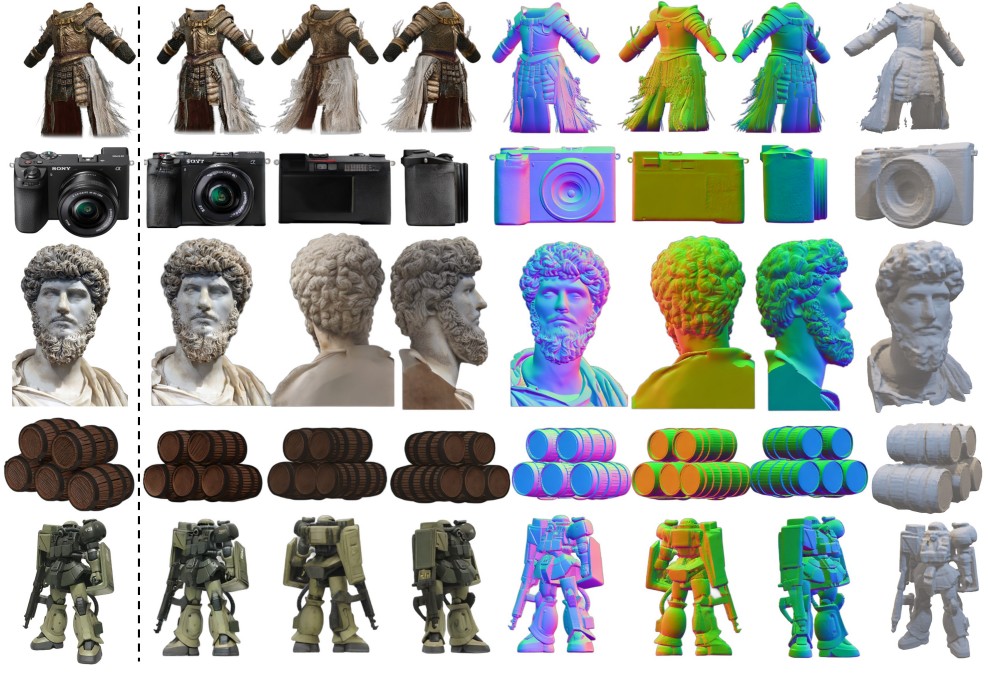

| Input images | Generated images | Generated normal maps | Output mesh |

Figure 1: Given single-view image with arbitrary intrinsic and viewpoints, Era3D can generate high-quality multiview images with a resolution of $512 \times 512$ on the orthogonal camera setting, which can be used in mesh reconstruction by NeuS [68].

To utilize image DMs for single-view 3D reconstruction, a pioneer work DreamFusion [46] tries to distill a 3D representation like NeRF [36] or Gaussian Splatting [62] from a 2D image diffusion by a Score Distillation Sampling (SDS) loss and many follow-up works improve the distillation-based methods in quality [71] and efficiency [62]. However, these methods suffer from unstable convergence and degenerated quality. Alternatively, recent works such as MVDream [55], SyncDreamer [33], Wonder3D [34] and Zero123++ [54] explicitly generate multiview images by multiview diffusion [64, 33] and then reconstruct 3D models from the generated images by neural reconstruction methods [68] or large reconstruction models (LRMs) [19, 27]. Explicitly generating multiview images makes these methods more controllable and efficient than SDS methods and thus is more popular in the single-view 3D reconstruction task.

Despite impressive advancements in multiview diffusion methods [55, 34, 33, 32, 31, 64, 63, 27], efficiently generating novel-view images for high-quality 3D reconstruction remains an open challenge. There are three noticeable challenges in the current approaches. (1) **Inconsistent predefined camera type**. Most multiview diffusion methods assume that the input images are captured by a camera with a predefined focal length. This leads to unwanted distortions when input images are captured by cameras with different camera types or intrinsics, as exemplified in Fig. 2 (e.g., Wonder3D's assumption of an orthogonal camera leads to distorted meshes when the input image is captured by a perspective camera with a small focal length). (2) **Inefficiency of multiview diffusion**. Multiview diffusion methods usually rely on multiview attention layers to exchange information among different views to generate multiview-consistent images. However, these multiview attention layers are usually implemented by extending the self-attention in Stable

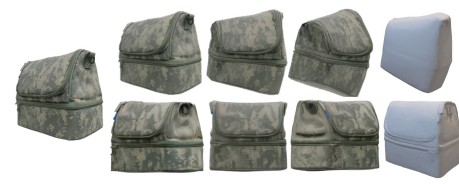

| Input | Generated images and mesh |

Figure 2: (top) Perspective input images for Wonder3D produce extreme distortion in the generation. (bottom) Era3D can handle images of commonly used intrinsics.

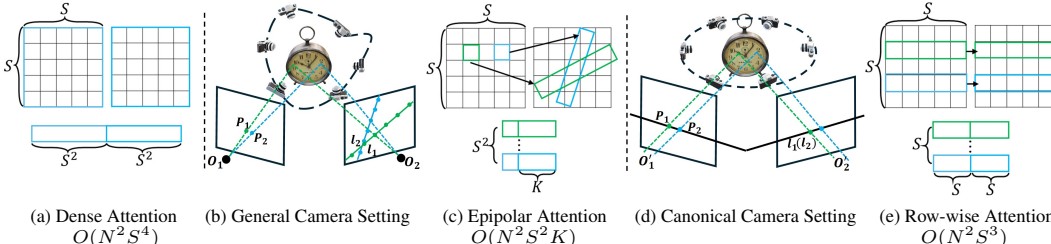

| (a) Dense Attention $O(N^2S^4)$ | (b) General Camera Setting | (c) Epipolar Attention $O(N^2S^2K)$ | (d) Canonical Camera Setting | (e) Row-wise Attention $O(N^2S^3)$ |

Figure 3: Different types of multiview attention layers. (**a**) In a dense multiview attention layer, all feature vectors of multiview images are fed into an attention block. For a general camera setting (**b**) with arbitrary viewpoints and intrinsics, utilizing epipolar constraint to construct an epipolar attention (**c**) needs to correlate the features on the epipolar line. This means that we need to sample $K$ points along each epipolar line to compute such an attention layer. In our canonical camera setting (**d**) with orthogonal cameras and viewpoints on an elevation of $0°$, epipolar lines align with the row of the images across different views (**e**), which eliminates the need to resample epipolar line to compute epipolar attention. We assume the latent feature map has a resolution of $H \times W$ and $H = W = S$. In such a $N$-view camera system, row-wise attention reduces the computational complexity to $O(N^2S^3)$.

Diffusion [52] to all multiview images, which is called dense multiview attention in Fig.3(a) and results in a significant increase in computation complexity and memory consumption. (3) **Low resolution of generated images**. The limitation above restricts most existing multiview diffusion models to resolutions of $256 \times 256$, preventing them from reconstructing detailed meshes. Addressing above challenges is crucial for developing practical and scalable multiview diffusion methods.

In this paper, we introduce Era3D, a novel multiview diffusion method that efficiently generates high-resolution ($512\times512$) consistent-multiview images for single-view 3D reconstruction. Unlike existing methods, Era3D allows images of commonly used camera types as inputs while mitigating the unwanted distortion brought by different camera models.

To this end, we employ a unique approach: using different camera models for input images and the generated ones for training, meaning that the input images are allowed to have arbitrary focal lengths and elevations while generated images are with orthogonal cameras and fixed viewpoints of $0°$ elevations. However, this requires DMs to implicitly infer and rectify the focal lengths and viewpoints of input images in the generation process, which is a challenging task and degrades the generation quality. To overcome this challenge and improve generation quality, we propose a novel regression and condition scheme and utilize the low-level feature maps of UNet at each denoising step to predict camera information of input images. We find that such a regression and condition scheme facilitates much more accurate camera pose prediction than existing methods [31] and leads to more details in the generation. As shown in Fig. 2, Era3D successfully avoids the above distortion problem brought by the different camera types and focal lengths.

Moreover, drawing inspiration from epipolar attention [65], Era3D enables efficient training for high-resolution multiview generation by introducing a novel row-wise multiview attention. Epipolar constraint can be utilized to constrain the attention regions across views and thus improve attention efficiency. However, directly applying such epipolar attention [65] for a general camera setting (Fig. 3(b)) is still memory and computationally inefficient because one would have to sample multiple points on epipolar lines for attention. This would require to construct a 3D grid of features in the view frustums for multiview images, which is too slow and memory-consuming. In contrast, since Era3D generates images with orthogonal cameras on viewpoints of $0°$, we find that epipolar lines in our camera setting are aligned with pixel rows of images across different views (Fig. 3(d)), which enables us to propose an efficient row-wise attention layer. Compared with dense multiview attention, row-wise attention significantly reduces memory consumption (35.32GB v.s. 1.66GB), and the computation complexity (220.41ms v.s. 2.23ms) of multiview attention (Fig. 3(e)). Even with Xformers [26], an accelerating library for attention, the efficiency of row-wise attention still outperforms existing methods by approximately twelve-fold as evident in Tab. 3. Consequently, the proposed row-wise attention allows us to easily scale Era3D to a high resolution of $512\times512$ to reconstruct more detailed 3D meshes.

Overall, our main contributions are summarized as follows: (1) Era3D is the first method that tries to solve the distortion artifacts brought by the inconsistent camera intrinsic in 3D generation; (2) we design a novel regression and condition scheme to enable diffusion models to take images of arbitrary cameras as inputs while outputting the orthogonal images on the canonical camera setting; (3) we propose row-wise multiview attention, an efficient attention layer for high-resolution multiview image generation; (4) our method achieves state-of-the-art performance for single-view 3D generation [10].

## 2   Related Works

Our study is primarily centered on the domain of image-to-3D. Unlike early works [6, 49, 62, 71, 29], which concentrate on per-scene optimization based on Score Distillation Sampling [46, 67], we emphasize feed-forward 3D generation.

**Image to 3D**. Generating 3D assets from images has been extensively researched, paralleling the development of GAN [13, 23, 38] and diffusion models (DMs) [57, 17, 11, 18]. A stream of these works [20, 79, 5, 19, 76], directly produce 3D representations, like SDF [43, 9, 41], NeRF [36, 14, 1], Triplane [3, 15, 56], Gaussian [24, 59] or 3D volume [60]. Zero-1-to-3 [32] and subsequent works [54, 31] represent the scene as a diffusion model conditioned on reference image and camera pose. LRM-based methods [19, 27, 70, 74] employ a large transformer architecture to train a triplane representation with a data-driven approach. Another technical line involves generating consistent multiview images first [55, 69, 54, 72], and then robustly reconstructing 3D shapes with NeuS [33, 34], Gaussian Splatting [61, 19, 75] or LRM [73]. Despite significant advancements, challenges remain in reconstruction quality, training resolution, or efficiency.

**Multiview diffusion**. Cross-view consistency is critical in 3D reconstruction and generation, relying on multiview feature correspondence to estimate 3D structures. MVffusion [64] first proposes generating multiview images in parallel with correspondence-aware attention, facilitating cross-view information interaction, and applies to texture scene meshes. [65, 22] introduces epipolar features into DM to enhance fusion between viewpoints. Zero123++ [54] tiles multi-views into a single image and performs a single pass for multiview generation, which is also used in Direct2.5 and Instant3D. MVDream [55] and Wonder3D [34] also design multiview self-attention to improve multiview consistency. Syncdreamer [33] composes multiview features into 3D volumes, conducting 3D-aware fusion in 3D noise space. All of the aforementioned methods share the same idea: modeling 3D generation with multiview joint probability distribution. Other works [8, 66] explore the priors from video diffusion model to achieve consistent multiview generation. Following widely used multiview self-attention, we propose more efficient row-wise attention to reduce computation workloads, but without loss of multiview feature interaction.

**Camera pose**. For 3D generation, early works [32, 55] are trained with fixed focal lens. When inference, one also needs to provide elevation of input for better performance. Further research seeks to mitigate this issue by leveraging fixed poses [34, 54] or incorporating additional elevation prediction modules [30]. LEAP [20] uses pixel-wise similarity, rather than estimated poses, to aggregate multiview features. However, none of these methods consider the distortion error caused by cameras, which can severely affect the reconstruction of real-world data. To overcome this, we opt to generate images at fixed views in the canonical orthogonal space, with simultaneous predictions of elevation and focal distortion.

## 3   Methods

Era3D is proposed to generate a 3D mesh from a single-view image. The overview is shown in Fig. 4, consisting of three key components. Given an input image with commonly used focal length and arbitrary viewpoint, Era3D generates multiview images in a canonical camera setting as introduced in Sec. 3.1. To improve the generation quality, in Sec. 3.2, we propose a regression and condition scheme, enabling diffusion models to predict accurate camera pose and focal lengths and guiding the denoising process. Finally, we considerably reduce memory consumption and improve computation efficiency by proposing row-wise multiview attention (Sec. 3.3), which exchanges information among multiview images to maintain multiview consistency. Finally, we reconstruct the 3D mesh from the generated images and normal maps using neural reconstruction methods like NeuS [68].

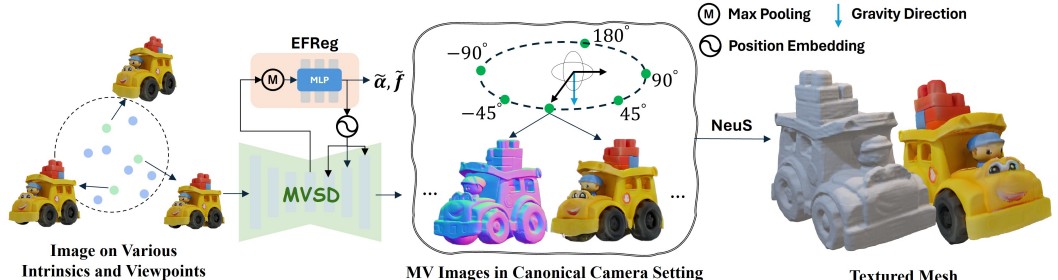

Figure 4: **Overview**. Given a single-view image as input, Era3D applies multiview diffusion to generate multiview consistent images and normal maps in the canonical camera setting, which enables us to reconstruct 3D meshes using NeuS [68, 37].

## 3.1 Camera Canonicalization

**Perspective distortion problem**. Existing multiview diffusion methods [33, 34, 54, 32] assume that the input image and the generated images share the same fixed intrinsic parameters. However, this assumption is often violated in practice, as input images may be captured by arbitrary cameras with varying focal lengths. When the input image has a different intrinsic matrix from the assumed one, the generated multiview images and reconstructed 3D meshes will exhibit perspective distortion, as illustrated in Fig. 2. This issue arises because these models are trained on renderings of the Objaverse [10] dataset with a fixed intrinsic and thus these models are severely biased towards the geometry patterns present in this fixed intrinsic matrix.

**Canonical camera setting**. To address this problem, Era3D uses different intrinsic parameters for input and generated images. Regardless of the focal length and pose of the input image, we consistently generate orthogonal images with an elevation of $0°$. For example, when processing an input image captured at elevation $\alpha$ and azimuth $\beta$, Era3D produces a set of multiview images at an azimuth of $\{\beta, \beta + 45°, \beta + 90°, \beta - 45°, \beta - 90°, \beta + 180°\}$ and an elevation of $0°$. We refer to the setup of these output images as a *Canonical* camera setting.

## 3.2 Regression and Condition Scheme

Given an image with an arbitrary viewpoint and focal length, generating novel-view images in the canonical camera setting is challenging because this implicitly puts an additional task on the diffusion model to infer the focal lengths and elevation of the input image. To make it easier, previous methods [32, 33, 2] rely on additional elevation inputs or predictions as the conditions for the diffusion model. However, pose estimation from a single image is inherently ill-posed due to the lack of geometry information. Moreover, even though estimating a rough elevation is possible, it is almost impossible for users to estimate the focal length of the input image.

To address this problem, we propose incorporating an Elevation and Focal length Regression module (**EFReg**) into the diffusion model. We use the feature maps of UNet to predict the camera pose in the diffusion process. Our motivation stems from the fact that the feature maps of UNet not only contain the input images but also include the current generation results which provide richer and more informative features for predicting the camera pose.

Specifically, within the middle-level transformer block of the UNet, we apply global average pooling to the hidden feature map $\mathbf{H}$, yielding a feature vector that is subsequently fed into three Multilayer Perceptron (MLP) layers $\mathcal{R}_1$ and $\mathcal{R}_2$ to regress the elevation $\tilde{\alpha}$ and focal lens $\tilde{f}$

$$\tilde{\alpha} = \mathcal{R}_1(\text{AvgPool}(\mathbf{H})), \tilde{f} = \mathcal{R}_2(\text{AvgPool}(\mathbf{H})). \tag{1}$$

The regressed elevation $\tilde{\alpha}$ and focal length $\tilde{f}$ are supervised by the ground-truth elevation $\alpha$ and focal length $f$ by

$$\ell_{\text{regress}} = \text{MSE}(\tilde{\alpha}, \alpha) + \text{MSE}(\tilde{f}, f). \tag{2}$$

Then, the regressed $\tilde{\alpha}$ and $\tilde{f}$ are used as conditions in the diffusion process. We apply positional encoding on $\tilde{\alpha}$ and $\tilde{f}$ and concatenate them with the time embeddings of the Stable Diffusion model.

The concatenated feature vectors are used in all the upsampling layers of the UNet, providing information about the estimated focal lengths and elevations for better denoising.

### 3.3 Row-wise Multiview Attention (RMA)

To generate multiview-consistent images, multiview diffusion models typically rely on multiview attention layers to exchange information among generated images. Such layers are often implemented by extending the existing self-attention layers of Stable Diffusion to conduct the attention on all the generated multiview images [34, 55, 69, 63]. However, this dense multiview attention can be computationally expensive and memory-intensive, as it processes all pixels of all multiview images. This limitation hinders the scalability of multiview diffusion models to high resolutions, such as $512 \times 512$.

Since the pixels of multiview images are related by epipolar geometry, considering epipolar lines in multiview attention could possibly reduce computational and memory complexity. However, strictly considering epipolar attention [65] in general two-view camera setting still consumes massive computation and memory. As shown in Fig. 3(b), for a pixel on camera $O_1$, we find its corresponding epipolar line in camera $O_2$ by the relative camera pose. Then, we need to sample $K$ points on the epipolar lines to conduct cross attention between these sample points of $O_2$ and the input pixel of $O_1$. In the following, we propose an efficient and compact row-wise multiview attention, which is a special epipolar attention tailored to our canonical camera setting.

In our canonical camera setting, cameras are distributed at an elevation of $0°$ around an object. Therefore, we can easily demonstrate the following proposition.

**Proposition 1** *If two orthogonal cameras look at the origin with their $y$ coordinate aligned with gravity direction and their elevations of $0°$ as shown in Fig. 3(d), then for a pixel with coordinate $(x, y) = (u, v)$ on one camera, its corresponding epipolar line on other views is $y = v$.*

We leave the proof in the supplementary material. Proposition 1 is a simplification of epipolar constraint, revealing that all epipolar lines correspond to rows in the generated multiview images. Building on this insight, we leverage the epipolar constraint by applying new self-attention layers on the same row across generated images to learn multiview consistency. By exploiting this constraint, we avoid the computational expense of dense multiview attention and instead accurately focus attention on epipolar lines. Moreover, our row-wise attention layer only involves elements from the same row, rather than sampling multiple points on the epipolar line, thereby significantly reducing computational complexity and facilitating training even on high-resolution inputs such as $512 \times 512$.

## 4   Experiments

**Datasets**. We trained Era3D on a subset of Objaverse [10]. To construct training images, we render 16 ground-truth images using orthogonal cameras with evenly distributed azimuth from $0°$ to $360°$ and a fixed elevation of $0°$. Subsequently, for each azimuth, we render 3 more images using perspective cameras and one image using an orthogonal camera, both of which have random elevations sampled from the range $[-20, 40]$ degrees. The perspective camera has a focal length randomly selected from the set $\{35, 50, 85, 105, 135\}$ mm, which are commonly used camera parameters. All the renderings have the resolution of $512 \times 512$. Following the previous methodologies [32, 33], we evaluate the performance of Era3D on the Google Scanned Object [12] dataset, widely regarded as a standard benchmark for 3D generation tasks. Moreover, we also evaluate our methods on in-the-wild images collected from the Internet or generated by image diffusion models [52] to show the generalization ability. The same as previous methods [55, 69], we remove backgrounds and center objects on these in-the-wild images.

**Metrics**. Our methodology is evaluated in two tasks, novel view synthesis (NVS) and 3D reconstruction. The NVS quality is evaluated by the Learned Perceptual Image Patch Similarity (LPIPS) [77] between the generated and ground-truth images. LPIPS evaluates the perceptual consistency because there may be slight misalignment between generated and ground-truth images. The 3D reconstruction quality is evaluated by the Chamfer Distance (CD) and the Volume IOU between the reconstructed meshes and the ground truth ones.

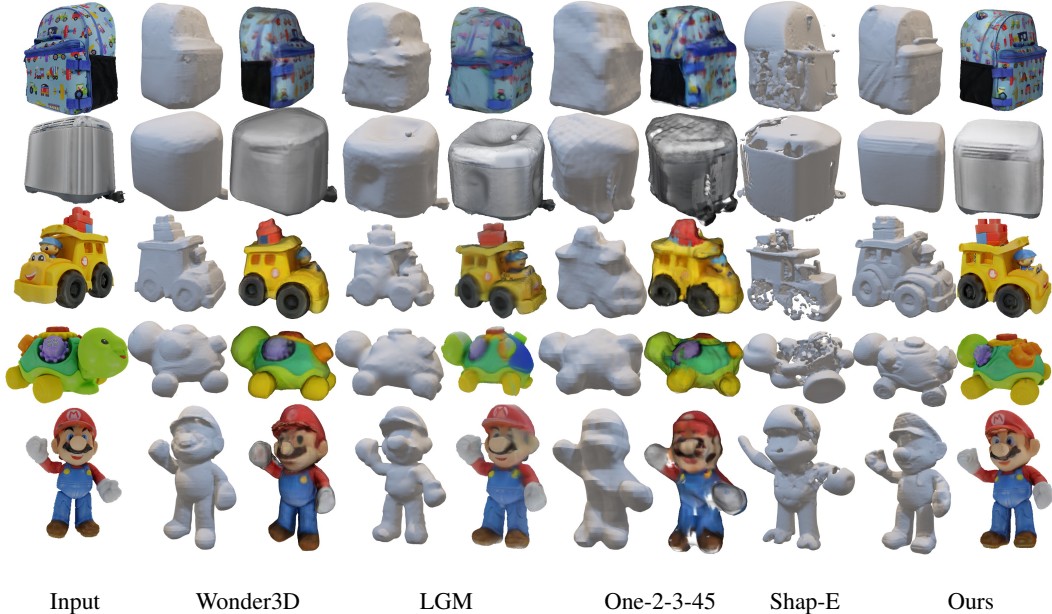

| Input | Wonder3D | LGM | One-2-3-45 | Shap-E | Ours |

Figure 5: Qualitative comparison of 3D reconstruction results on the GSO dataset [12]. Era3D produces the most high-quality 3D meshes with more details than baseline methods.

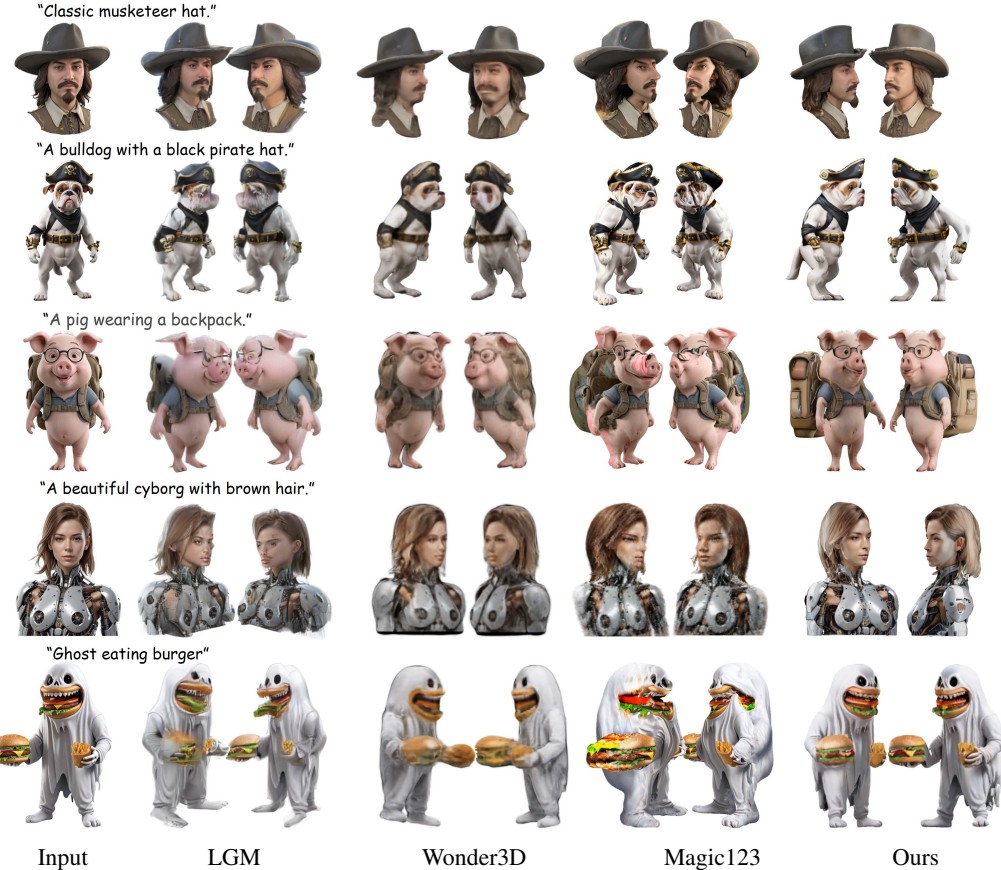

"Classic musketeer hat."

"A bulldog with a black pirate hat."

"A pig wearing a backpack."

"A beautiful cyborg with brown hair."

"Ghost eating burger"

| Input | LGM | Wonder3D | Magic123 | Ours |

Figure 6: Qualitative comparisons of novel view synthesis quality of reconstructed 3D meshes with single-view images generated by SDXL [45].

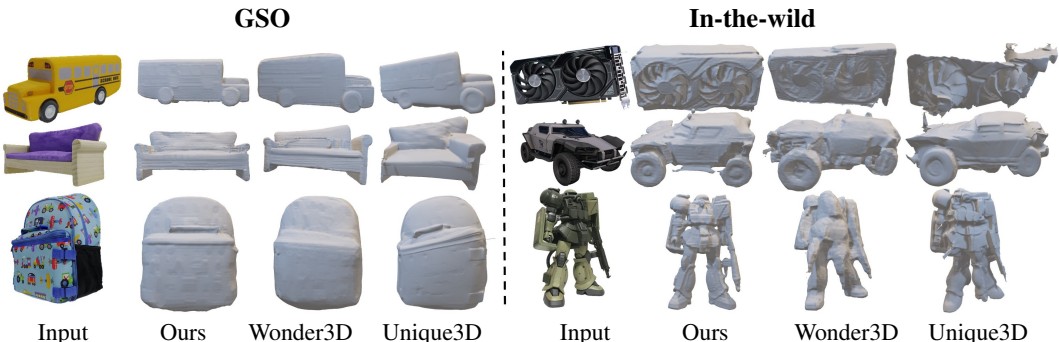

| **GSO** | | | | **In-the-wild** | | | |
| Input | Ours | Wonder3D | Unique3D | Input | Ours | Wonder3D | Unique3D |

Figure 7: More comparisons w.r.t distortion problem.

**Implementation details**. Our implementation is built upon the open-source text-to-image model, SD2.1-unclip [51]. We train Era3D on 16 H800 GPUs (each with 80 GB) using a batch size of 128 for 40,000 steps. We set the initial learning rate as 1e-4 and decreased it to 5e-5 after 5,000 steps. The training process takes approximately 30 hours. To conduct classifier-free guidance (CFG) [18], we randomly omit the clip condition at a rate of 0.05. During inference, we employ the DDIM sampler [57] with 40 steps and a CFG scale of 3.0 for the generation. Following Wonder3D [34], we first perform reconstruction with NeuS and then apply a texture refinement step. The whole pipeline requires approximately 4 minutes, comprising 13 seconds for the multiview diffusion, 3 minutes for the NeuS reconstruction, and 10 seconds for the texture refinement. For further details, please refer to the supplementary material.

## 4.1 Experimental Results

**Novel view synthesis**. First, several examples of the multiview images and normal maps generated by Era3D are shown in Fig. 1. The results demonstrate that given input images with varying focal lengths and viewpoints, Era3D can generate high-quality and consistent multiview images and normal maps. When the input image is captured by a perspective camera and its viewpoint is not on an elevation of $0°$, Era3D can correctly perceive the elevation of the viewpoint and the perspective distortion. Then, our method learns to generate images of the same object with high fidelity using orthogonal cameras on canonical viewpoints, effectively reducing the artifacts caused by the perspective distortion and improving the reconstruction quality. Moreover, Era3D can produce images in the $512 \times 512$ resolution, which enables generating much more details like the fine-grained texture on the "Armor" and the complex structures on the "Mecha" in Fig. 1.

Furthermore, we provide a quantitative comparison with other single-view reconstruction methods including RealFusion [35], Zero-1-to-3 [32], SyncDreamer [33] and Wonder3D [34] in Tab. 1. The results show that our method outperforms previous approaches in terms of novel-view-synthesis quality by a significant margin, showcasing the effectiveness of our designs.

**Reconstruction**. We further conduct experiments to evaluate the quality of reconstructed 3D meshes. We compare our method with RealFusion [35], Zero-1-to-3 [32], One-2-3-45 [31], Shap-E [21], Magic123[49], Wonder3D [34], SyncDreamer [33], and LGM [61]. Reconstructed meshes and their textures on the GSO dataset are shown in Fig. 5 while the renderings of the reconstructed meshes on text-generated images are shown in Fig. 6. As shown in the results, Shap-E fails to generate completed structures. The meshes reconstructed by One-2-3-45 and LGM tend to be over-smoothed and lack details due to the multiview inconsistency in generated images by Zero-1-to-3 [32] or ImageDream [69]. The results of Wonder3D tend to be distorted on these input images rendered with a focal length of 35mm because it assumes the input images are captured by orthogonal cameras. In contrast, our results show significant improvements in terms of completeness and details than these baseline methods.

Quantitative comparisons of Chamfer Distance (CD) and Intersection over Union (IoU) are shown in Tab. 1. Era3D outperforms all other methods, exhibiting lower Chamfer Distance and higher Volume IoU, suggesting that the meshes it generates align more closely with the actual 3D models.

**Distortion problem** We finally provide the comparisons with SOTA methods, like Unique3D, w.r.t distortion problem in Fig. 7, which shows that they suffers from severe perspective distortion while our method greatly alleviates this problem.

Table 1: Quantitative evaluation of Chamfer distance, IoU (for reconstruction), and LPIPS (for NVS).

| Method | CD ↓ | IoU ↑ | LPIPS ↓ | SSIM ↑ | PSNR ↑ |
|---|---|---|---|---|---|
| RealFusion | 0.0819 | 0.2714 | 0.283 | 0.722 | 15.26 |
| Zero-1-to-3 | 0.0339 | 0.5035 | 0.166 | 0.779 | 18.93 |
| One-2-3-45 | 0.0629 | 0.4086 | - | - | - |
| Shap-E | 0.0436 | 0.3584 | - | - | - |
| Magic123 | 0.0516 | 0.4528 | - | - | - |
| SyncDreamer | 0.0261 | 0.5421 | 0.146 | 0.798 | 20.05 |
| Wonder3D | 0.0248 | 0.5678 | 0.141 | 0.811 | 20.83 |
| LGM | 0.0259 | 0.5628 | - | - | - |
| Ours | **0.0217** | **0.5973** | **0.126** | **0.837** | **22.74** |

Table 2: Comparison of pose estimation accuracy. $\tilde{\alpha}$: elevation, $\tilde{f}$: normlized focal length.

| | Method | $\tilde{\alpha}$ / ° | $\tilde{f}$ / mm |
|---|---|---|---|
| | Dino | 10.24 | 0.28 |
| Error | One-2-3-45 | 10.14 | - |
| | Ours | **2.69** | **0.13** |
| | Dino | 377.07 | 0.058 |
| Variance | One-2-3-45 | 267.44 | - |
| | Ours | **112.30** | **0.036** |

## 4.2 Accuracy of Estimated Elevations and Focal Lengths

Beyond the tasks already discussed, we further evaluate the pose prediction of Era3D on the GSO dataset. We render the images with an elevation of $[-10, 40]$ degrees and focal lengths of $\{35, 50, 85, 105, 135, \infty\}$, respectively. As a baseline method, we employ dinov2_vitb14 feature [40] to predict the pose and train it with the same dataset. We compare our predictions with this baseline method and One-2-3-45. As shown in Tab. 2, Era3D achieves superior performance in error and variance. A more detailed analysis is provided in the supplementary materials.

## 4.3 Ablations and Discussions

**Regression and condition scheme**. In Fig. 8, we remove the EFReg and compare the results with our full model to demonstrate the effectiveness of our design. Without regressing a focal length and elevation as conditions during the denoising process, the resulting shape is distorted and fails to generate reasonable novel views in the canonical camera setting. In comparison, adding our EFReg module and conditioning on the predicted elevations and focal lengths provides effective guidance to generate undistorted cross-domain images, thereby resulting in more accurate 3D reconstructions.

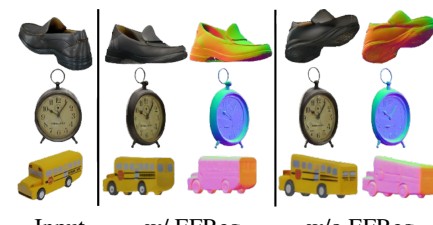

Input     w/ EFReg     w/o EFReg

Figure 8: Ablation study of EFReg.

**Row-wise multiview attention**. As illustrated in Fig. 1, our proposed RMA effectively facilitates information exchange among multiview images, yielding consistent results comparable to those achieved by dense multiview attention layers in [69, 34]. In a $N$-view camera system, assuming a latent feature with size of $S \times S$, our RMA design significantly improves training efficiency by reducing the computational complexity of attention layers from $O(N^2S^4)$ to $O(N^2S^3)$, as shown in Fig. 3. While epipolar attention also achieves a complexity reduction to $O(N^2S^2K)$, where $K$ is the sample number, it does so at the cost of increased memory and time consumption due to the sampling process. To further highlight the efficiency of RMA over dense multiview attention, we present the memory usage and running time of both 256 and 512 resolutions. We use the epipolar attention implementation in [22]. As listed in Tab. 3, the advantage of RMA becomes increasingly obvious as the resolution grows. At a resolution of 512, RMA achieves a thirty-fold reduction in memory usage and a nearly hundred-fold reduction in running time. Even with xFormers [26], our method substantially improves training efficiency by a large margin (22.9 ms vs. 1.86 ms). This efficiency enables training models on higher resolutions or with denser views without significantly increasing computational efficiency and demand, thereby maintaining a lightweight framework.

Finally, we conduct performance comparisons between dense, epipolar, and our row-wise attention. Considering that in our orthogonal setup with the same elevation, epipolar attention is equivalent to row-wise attention (except for implementation differences), we compared dense and row-wise attention at a resolution of 256. Era3D achieves comparable performance with dense multiview attention, as evidenced in Tab. 4. Our row-wise setting outperforms the baseline for the Chamfer

Table 3: Memory usage and running time of multiview attention with resolution of 256 and 512.

| Multiview attention | w/o xFormers | | | | w/ xFormers | | | |
| --- | --- | --- | --- | --- | --- | --- | --- | --- |
| | Memory usage (G) | | Running time (ms) | | Memory usage (G) | | Running time (ms) | |
| | 256 | 512 | 256 | 512 | 256 | 512 | 256 | 512 |
| Dense | 3.02 | 35.32 | 8.88 | 220.41 | **0.99** | 1.42 | 1.77 | 22.96 |
| Epipolar | 2.43 | 24.20 | 3.57 | 60.89 | 1.02 | 1.71 | 1.78 | 20.03 |
| Row-wise | **0.95** | **1.66** | **0.91** | **2.23** | **0.99** | **1.08** | **0.28** | **1.86** |

Table 4: Performance of dense and row-wise attention at a resolution of 256.

| Method | CD ↓ | IoU ↑ | LPIPS ↓ | PSNR ↑ | SSIM ↑ |
| --- | --- | --- | --- | --- | --- |
| dense | 0.0239 | 0.5877 | 0.140 | 20.73 | 0.819 |
| rowwise | 0.0232 | 0.5831 | 0.137 | 20.92 | 0.813 |

distance, LPIPS, and PSNR. We attribute this to row-wise attention reducing the number of attention tokens, allowing the model to focus more on valuable tokens.

## 5 Limitation and Conclusion

**Limitations**. Though Era3D achieves improvements on the multiview generation task, our method struggles to generate intricate geometries like thin structures because we only generate 6 multiview images and such sparse generated images have difficulty in modeling complex geometries. Since the reconstruction algorithm is based on Neural SDF, Era3D cannot reconstruct meshes with open surfaces. In future works, we could integrate our framework with other 3D representations, such as Gaussian splatting, to improve both rendering and geometry quality.

**Conclusion**. In this paper, we present Era3D, a high-quality multiview generation method for single-view 3D reconstruction. In Era3D, we propose to generate images in a canonical camera setting while allowing input images to have arbitrary camera intrinsics and viewpoints. To improve the generation quality, we design a regression and condition scheme to predict the focal length and elevation of input images, which are further conditioned to the diffusion process. Additionally, we employ row-wise multiview attention to replace dense attention, significantly reducing computational workloads and facilitating high-resolution cross-view generation. Compared with baseline methods, Era3D achieves superior geometry quality in single-view 3D reconstruction.

## 6 Ethics Statement

The objective of Era3D is to equip users with a powerful tool for creating detailed 3D models. Our method allows users to generate 3D objects based on a single image. However, there is a potential risk that these generated models could be misused to deceive viewers. It is important to note that this issue is not unique to our methodology but prevalent in other generative model methodologies. Therefore, it is absolutely essential for current and future research in the field of 3D generative modeling to address and reassess these considerations consistently.

## Acknowledgments and Disclosure of Funding

The research was supported by Theme-based Research Scheme (T45-205/21-N) from Hong Kong RGC, and Generative AI Research and Development Centre from InnoHK.

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

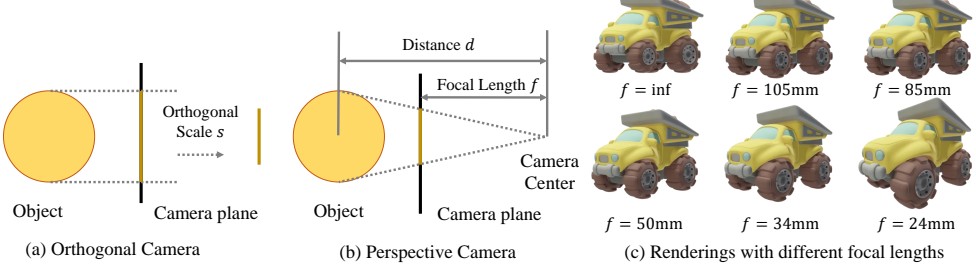

(a) Orthogonal Camera      (b) Perspective Camera      (c) Renderings with different focal lengths

Figure 9: Equivalence between orthogonal and perspective camera models and our rendering samples in GSO dataset [12].

# A    Supplementary Material

## A.1    Implementation Details

**Data preparation**. To render training data, we randomly choose a focal length for the input image from a predefined set of focal lengths and we always use the same orthogonal camera for all generated images. To train Era3D, we need to construct an orthogonal camera to render generation images and the equivalent perspective cameras to render input images. The equivalence here means that the renderings from these two kinds of cameras have almost the same size, for reducing training bias. As shown in Fig. 9 (a) and (b), given a predefined orthogonal scale $s$ and a given focal length $f$, we compute the distance $d$ by $d = f/s$, where $s$ is the orthogonal scale to scale the size of the rendered image. We will adjust the distance from the camera center to the object to the value $d$ to make the rendered image as similar as possible.

**Normalized focal length**. In our experiment, we choose several discrete focal lengths, $\{24, 35, 50, 85, 105, 135\}$ mm, and an orthogonal camera to render multiview images for training. Due to the broad range of focal lengths, we regress the normalized focal lengths instead of the actual ones. Specifically, we normalize them with minimum focal,

$$\tilde{f} = 24/f. \tag{3}$$

For the orthogonal views, we set $\tilde{f} = 0$.

**Cross-domain attention**. In contrast to Wonder3D [34], we do not incorporate an additional attention layer to fuse information between cross-domain images. Instead, in earlier experiments, we combined both the cross-domain fusion module and the multiview fusion module within the same row-wise attention block to enhance training efficiency. This initial setup, however, leads to suboptimal outcomes, as shown in Fig. 5, where it tends to produce overly smoothed normal maps. We hypothesize that the significant differences between domain images are the primary cause of this issue, as the row-wise information alone is insufficient for learning fine-grained features across domains.

In contrast, integrating the cross-domain module within the self-attention block enables the utilization of complete image features, allowing the model to better handle these disparities and achieve improved generalization. Our attention block consists of self-cross-domain attention, row-wise multiview attention, and cross-domain attention, which allows us to merge the two-stage training into a single-stage joint training, effectively aligning images of both domains.

**Reconstruction and texture refinement**. We perform reconstruction from the generated multiview normal and color images using NeuS. However, since we only generate sparse views, the rendered images of the reconstructed meshes may not attain the quality of the generated ones, particularly for objects with complex textures. To address this issue, we employ differential rendering to refine the textures on the reconstructed meshes.

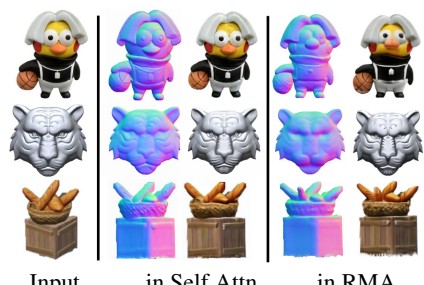

Input     in Self Attn.     in RMA

Table 5: Ablation study on the position of the cross domain-block.

Specifically, we preserve the geometry of Neus and initialize the texture with vertex colors. We utilize [25] to render multiview images in predefined views and optimize the vertex colors with the corresponding generated images. This process takes less than 1 second and significantly improves the texture quality. For the NeuS reconstruction, we use the same settings as Wonder3D. During texture refinement, we optimize the appearance for 200 iterations with a resolution of 1024 and a learning rate of 1e-3.

**Noise scheduler**. The original SD2.1-Unclip base model employs a scaled linear noise scheduler during training, which is effective for smaller sample sizes, such as 256. However, this scheduler tends to restrict the generative capabilities of models at larger sizes, such as 512. Drawing inspiration from recent research by Chen [7], we implemented a linear noise scheduler throughout our experiments to enhance generation quality and accelerate the convergence of the background. This adjustment proves critical in supporting the model's performance and efficiency.

## A.2 Proof of Proposition 1

Given two cameras $O_1$ and $O_2$ with relative rotation $R$ and translation $t$, let $x_1$ and $x_2$ denote the homogeneous coordinates of corresponding points in the image planes, respectively, epipolar constraint can be expressed as

$$x_2^T E x_1 = 0, \tag{4}$$

where $E$ is the fundamental matrix. Then, the epipolar line in $O_2$ can be represented as

$$l = E x_1, E = [t]_\times R, \tag{5}$$

where $[t]_\times$ denotes the skew-symmetric matrix generated by $t$. However, epipolar attention needs to query dense points on epipolar lines, which leads to significant computational inefficiency, due to arbitrary directions and varying lengths of epipolar lines.

Row-wise multiview attention is a special epipolar attention in our defined canonical camera setting, as depicted in Fig. 3(d). Assuming the y-axis represents the gravity direction, the x-axis denotes the right-hand direction, and the z-axis points away from the camera to the origin of the object. The relative $R$ and $t$ between two cameras can be represented as

$$R = \begin{bmatrix} cos(\theta) & 0 & sin(\theta) \\ 0 & 1 & 0 \\ -sin(\theta) & 0 & cos(\theta) \end{bmatrix}, t = [t_x, 0, t_z]^T, \tag{6}$$

where $\theta$ is the azimuth angle of the cameras. Considering the point $P_1(x_1, y_1, z_1)$ in camera $O_1'$, the coordinates of $P_1$ in camera $O_2'$ are given by

$$P_2(x_2, y_2, z_2) = R P_1 + t = \begin{bmatrix} cos(\theta)x + sin(\theta)z_1 + t_x \\ y_1 \\ cos(\theta)z - sin(\theta)x_1 + t_z \end{bmatrix}. \tag{7}$$

It is observed that the scene points from different views share the same y-coordinate. Assuming the identical orthographic scale, the y-coordinates of projections on the image plane are also equivalent. Extending a single sample to a line of points with the same y-coordinate, all projections of those points on two image planes are on the same row.

## A.3 Pose Estimation

During inference, we obtain the final pose by averaging the class-free guidance results from all denoising steps,

$$\tilde{\alpha} = \frac{1}{T} \sum_{t=1}^{T} [(1+w)\alpha_\theta^t(z, c) - w\alpha_\theta^t(z)],$$

$$\tilde{f} = \frac{1}{T} \sum_{t=1}^{T} [(1+w)f_\theta^t(z, c) - wf_\theta^t(z)], \tag{8}$$

where $z$ is the latent, $c$ is the condition, $w$ is the CFG scale, $\theta$ is the parameters of UNet and MLP and $T$ is the denoising step. We observe that the averaged class-free guidance predictions achieve the highest accuracy. We attribute this to the random dropping of condition images during training. We do not rely solely on the prediction at the final step because we constrain the estimated elevation and focal length with the ground pose rather than the noisy ones at each denoising step. We conduct extensive experiments to evaluate the accuracy of the estimated pose on the GSO dataset.

**Elevation**. As listed in Tab. 6, Dino and One-2-3-45 only learn the relative trend as the elevation increases, resulting in large errors for cases with high elevations. This is because our baseline only employs the feature of a single image, which is insufficient for predicting the absolute elevation. The prediction of One-2-3-45 mainly depends on the multiview images generated by Zero-1-to-3. The inconsistency of generated images leads to ambiguity in prediction. In comparison, our results are closer to the ground truth elevation. In most cases, the variance of our method is significantly lower than that of Dino and One-2-3-45, demonstrating remarkable robustness.

**Focal**. We report the error and variance of normalized focal predictions for the baseline and our method in Tab. 7. It is observed that the baseline cannot distinguish the differences between various focal lengths. In contrast, our

Table 6: Elevation accuracy on GSO dataset.

| | Method | Elevation / ° | | | | | | | | | | | |
| | | -10 | | 0 | | 10 | | 20 | | 30 | | 40 | |
| | | Pred | Err | Pred | Err | Pred | Err | Pred | Err | Pred | Err | Pred | Err |
| Mean | Dino | -13.58 | 3.58 | -6.83 | 6.83 | 4.63 | 5.37 | 10.21 | 9,79 | 16.72 | 14.28 | 18.41 | 21.59 |
| | One-2-3-45 | -11.93 | 1.93 | -5.23 | 5.23 | -0.06 | 10.06 | 11.17 | 8.83 | 15.5 | 14.5 | 19.73 | 20.27 |
| | Ours | -9.71 | **0.29** | -2.20 | **2.20** | 5.59 | **4.41** | 23.85 | **3.85** | 34.03 | **4.03** | 38.67 | **1.33** |
| Var | Dino | 202.37 | - | 252.51 | - | **153.67** | - | 365.32 | - | 463.92 | - | 824.63 | - |
| | One-2-3-45 | 175.02 | - | 103.64 | - | 168.61 | - | 200.97 | - | 326.05 | - | 630.32 | - |
| | Ours | **32.1** | - | **83.63** | - | 163.38 | - | **113.55** | - | **134.69** | - | **146.46** | - |

Table 7: Focal length accuracy on GSO dataset.

| | Method | Normalized focal length (focal length / mm) | | | | | | | | | | | |
| | | 0.68 (35) | | 0.48 (50) | | 0.28 (85) | | 0.22 (105) | | 0.17 (135) | | 0.0 (ortho) | |
| | | Pred | Err | Pred | Err | Pred | Err | Pred | Err | Pred | Err | Pred | Err |
| Mean | Dino | 0.75 | 0.07 | 0.46 | 0.02 | 0.53 | 0.27 | 0.56 | 0.34 | 0.57 | 0.4 | 0.59 | 0.59 |
| | Ours | 0.69 | 0.01 | 0.54 | 0.06 | 0.37 | 0.09 | 0.33 | 0.11 | 0.34 | 0.17 | 0.32 | 0.32 |
| var | Dino | 0.069 | - | 0.073 | - | 0.054 | - | 0.049 | - | 0.052 | - | 0.049 | - |
| | Ours | 0.041 | - | 0.035 | - | 0.024 | - | 0.02 | - | 0.069 | - | 0.032 | - |

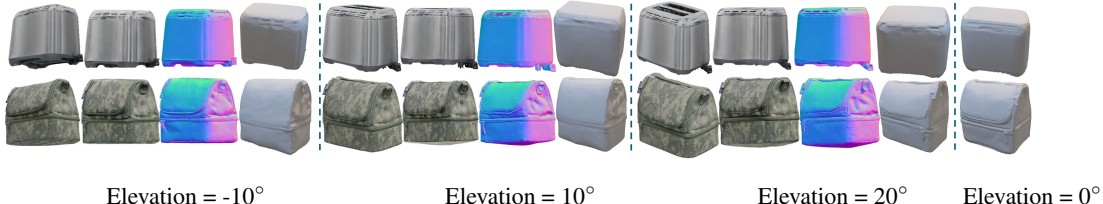

Elevation = -10°         Elevation = 10°         Elevation = 20°     Elevation = 0°

Figure 10: Generation results of various elevation. We use reconstructions from the view of Elevation = 0° as the reference. Different inputs generate consistent results.

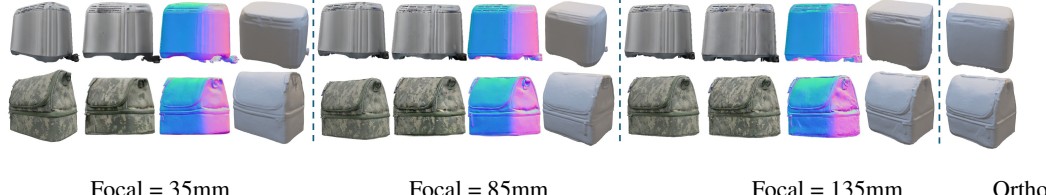

Focal = 35mm         Focal = 85mm         Focal = 135mm     Ortho

Figure 11: Generation results of various distortions. We employ reconstructions from orthogonal inputs as the reference. For each case, we illustrate the input along with the generated color, normal, and mesh.

Table 8: Quantitative evaluation on images with various focal lengths. CD: Chamfer Distance.

| Pose | $\alpha=0$ | | | | | $f=\infty$ | | | | | $\alpha=0$ |
|------|------------|--------|--------|---------|---------|-------------|-------------|-------------|-------------|-------------|------------------|
| | $f$=35 | $f$=50 | $f$=85 | $f$=105 | $f$=135 | $\alpha$=-10 | $\alpha$=10 | $\alpha$=20 | $\alpha$=30 | $\alpha$=40 | $f = \infty$ |
| CD | 0.0223 | 0.0219 | 0.0216 | 0.0214 | 0.0214 | 0.0217 | 0.0216 | 0.216 | 0.0219 | 0.0217 | 0.0213 |

model can predict the large distortion (e.g., focal=35, 50) of input images, which is advantageous for correcting them to some extent. Simultaneously, our method exhibits smaller errors for settings with large focal lengths than the baseline. We believe our method could be further explored and improved in the future.

Furthermore, we use orthogonal renderings at an elevation of 0 degree as the reference. We vary the elevation from $-10$ to $40$ degrees and select focal lengths from $\{35, 50, 85, 105, 135, \infty\}$ to assess the system's robustness to elevations and focal distortions by reconstructed meshes. As indicated in Tab. 8, the orthogonal setting at elevation 0 achieves the best performance of CD, and other settings are on the same bar with the reference setting, suggesting that our method effectively handles these distortions, producing meshes that align well with the reference ones, even under significant focal distortion. We visualize some samples in Fig. 10 and Fig. 11.

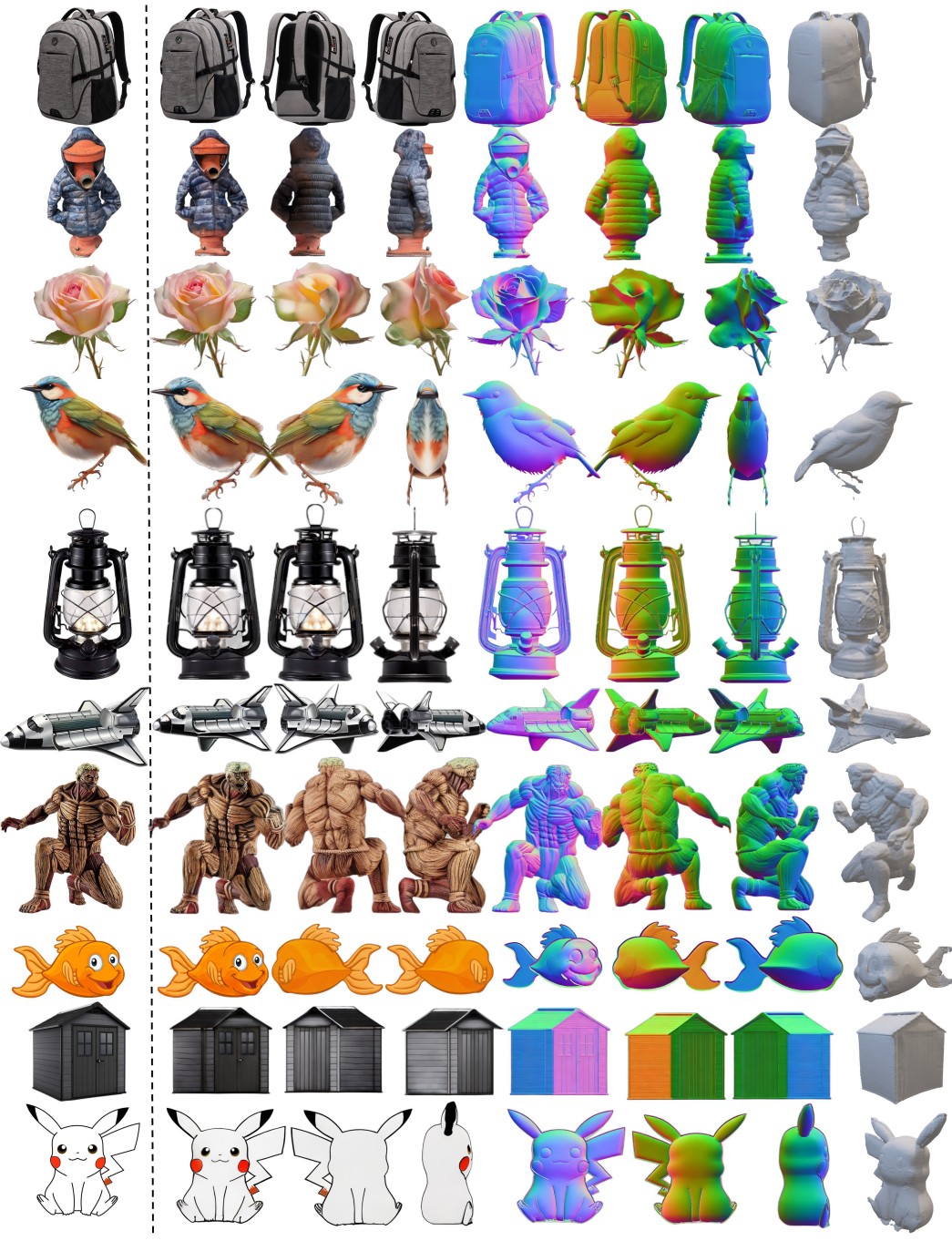

Figure 12: More results of images from the Internet.

