# OpenReview forum: "Era3D: High-Resolution Multiview Diffusion using Efficient Row-wise Attention"
_NeurIPS.cc/2024/Conference — NeurIPS 2024 poster_

### Official Review · Reviewer_aLmz · 2024-07-16

**Soundness:** 3
**Presentation:** 2
**Contribution:** 3
**Rating:** 7
**Confidence:** 4

**Summary:**

This paper develops a new image-to-multiview diffusion model with two key highlights. First, it proposes a novel method for estimating the focal length and elevation. Second, it introduces a new cross-view attention mechanism. Experiments demonstrate that the method outperforms previous SOTAs.

**Strengths:**

Previous image-to-3D methods struggle when the object in the image is distorted due to a small focal length. Therefore, this paper focuses on a practical problem. Given the development of previous image-to-multiview methods, it is common to apply a cross-view attention mechanism to address the multiview consistency problem. This paper reconsiders the shortcomings of previous cross-view attention methods and proposes a better solution.

**Weaknesses:**

My main concern is with the presentation of experiments. As mentioned in L136, the motivation is to improve performance on real-world data, where small focal-length cases exist. Fig. 2 shows a good example. Before reading the experiments, I expected to see more results like Fig. 2. However, I think only the toaster example in Fig. 5 demonstrates the superiority of Era3D in this regard. The main paper should present more results.

By comparison, the results in Fig. 6 seem more focused on demonstrating the generalization ability of Era3D and how other methods suffer from out-of-distribution issues. Although I understand that the robust generalization ability of Era3D might be due to EFReg, given that other methods are also trained on Objaverse, I think this presentation is not explicit. In particular, the focal length of the images output by SDXL cannot be controlled, so it is unclear whether Era3D's better performance is due to EFReg or better preprocessing of the training data.

Moreover, I disagree with the claim in L97 that Era3D is the first method to solve distortion artifacts. For example, "Shape, Pose, and Appearance from a Single Image via Bootstrapped Radiance Field Inversion" (CVPR23) also explores this issue by training directly on Pscal3D+, a real-world dataset with varying focal lengths. VQA-Diff (ECCV24) explores a similar problem and proposes to utilize prior knowledge of LLMs. Despite this, I find the proposed EFReg to be an interesting and novel solution.

**Questions:**

(1) I think Wonder3D is a highly related paper, as both deal with the image-to-multiview problem. Given that the authors claim one of Era3D's superiorities is its high resolution, I'm wondering why they did not use PSNR and SSIM to evaluate the synthetic novel views as Wonder3D did.
(2) As explained by the authors in L207, the RMA is based on a simplification. Table 3 shows the benefits of RMA. However, my intuition is that RMA sacrifices quality to some degree compared to Dense and Epipolar. I did not find an ablation study regarding this in the main paper or supplementary material. I think the authors should demonstrate that the quality of RMA, Dense/Epipolar are comparable. For example, if the quality deterioration of RMA exists, it should be shown to be acceptable. Otherwise, the authors should demonstrate that RMA outperforms Dense and Epipolar in terms of quality and explain the reason.

**Limitations:**

This paper addresses the issue of varying focal lengths in real-world data, which is a good motivation in my view. However, I do not think Era3D fully closed this problem, given that only a limited number of focal lengths are considered when preparing the training data. The authors should include this as a limitation or discuss it as a possible future work.

---

> ### Author Rebuttal · Authors · 2024-08-06
>
> Thank you for your valuable time and insightful comments! We have tried to address your concerns in the updated manuscript and our rebuttal text:
>
> **Q1: More results of perspective input.**
> We appreciate this suggestion and have accordingly expanded our results section. In Figure 2 of the global response,  we include more results on GSO datasets and cases from the Internet. Compared with Wonder3D and more recent Unique3D, our method demonstrates superior performance in mitigating distortion and reconstructing plausible 3D shapes. As per your recommendation, we will incorporate more results into the main paper.
>
> **Q2: The presentation about EFReg is explicit. Is the better performance from EFReg or preprocessing of training data?**
> In Fig.7 of the main paper, we evaluate the effectiveness via qualitative comparison by removing EFReg. Notably, without EFReg, the resulting shape is distorted and
> fails to generate reasonable novel views in the canonical camera setting. To provide quantitative support for these observations, we additionally report the Chamfer Distance in Tab.1. We use orthogonal renderings at an elevation of 0° as the reference (last column) and vary the elevation from −10° to 40° and select focal lengths from {35, 50, 85, 105, 135, ∞} to assess the system’s robustness to distortions. These results consistently confirm that EFReg significantly contributes to robust pose estimation and enhances overall reconstruction accuracy.
>
> **Table 1**: Ablation of EFReg on GSO datasets with various elevation (α) and focal lengths (f). We report the Chamfer Distance (↓).
>
> | Pose | α=0 | | | | | f=∞ | | | | | α=0, f=∞ |
> |------|-----|-----|-----|-----|-----|-----|-----|-----|-----|-----|------------|
> |      | f=35 | f=50 | f=85 | f=105 | f=135 | α=-10 | α=10 | α=20 | α=30 | α=40 | |
> | w/o EFReg | 0.0237 | 0.0233 | 0.020 | 0.022 | 0.0217 | 0.0221 | 0.0217 | 0.0225 | 0.0231 | 0.0228 | 0.0217 |
> | w EFReg | 0.0223 | 0.0219 | 0.0216 | 0.0214 | 0.0214 | 0.0217 | 0.0216 | 0.0216 | 0.0219 | 0.0217 | 0.0213 |
>
> **Q3: Missing metrics of PSNR and SSIM.**
> We apologize for the missing reports of PSNR and SSIM. Following your valuable suggestion, we have now incorporated these results in Tab.2 to provide a more comprehensive evaluation following your suggestion. Our method significantly outperforms others, consistent with our performance on other geometry metrics.
>
> **Table 2:** Quantitative evaluation of SSIM and PSNR.
> | Method | RealFusion | Zero-1-to-3 | SyncDreamer | Wonder3D | Ours |
> |--------|------------|-------------|-------------|----------|------|
> | SSIM(↑) | 0.722 | 0.779 | 0.798 | 0.811 | **0.837** |
> | PSNR(↑) | 15.26 | 18.93 | 20.05 | 20.83 | **22.74** |
>
>
> **Q4: Inaccurate claim of 'the first method to solve perspective distortion'.**
> We acknowledge that our initial statement requires refinement. The study in CVPR'23 attempts to predict the distortion in category-specific image reconstruction. VQA-Diff only explores vehicle distortion in Autonomous Driving. In contrast to them, our study first considers distortion artifacts for general 3D object generation. We will revise the statement in the manuscript.
>
> **Q5: Does RMA lead to performance degradation compared with dense/epipolar MV attention?**
> Theoretically, row-wise, epipolar, and dense MV attentions are equivalent in our orthogonal setup. However, both dense MV attention and epipolar attention consume a large amount of GPU memory in the training, which cannot scale up to high resolution 512$\times$512 and largely reduces the training batch size. As observed by Zero123 and other works, a large training batch size is very important for training a high-quality diffusion model. Due to the memory limitation, we cannot conduct a fair comparison experiment to train the model with dense MV attention or epipolar line attention.
>
> **Q6: Era3D does not fully address the distortion issue. It should be included as a limitation.**
> Thanks for your kind reminder. We use the commonly used focus lengths for training Era3D, which significantly mitigates perspective distortions. We acknowledge that our work offers a promising avenue for addressing this challenge rather than fully resolving the issue. We will clarify this point and discuss it in the limitation section.

---

> > ### Comment · Reviewer_aLmz · 2024-08-08
> > **response to rebuttal**
> >
> > The authors addressed most of my concerns. However, I believe there is still a limitation/weakness remaining.
> >
> > [Strength and Suggestions]
> >
> > Fig. 2 looks good, and I think it provides a more meaningful comparison than the current Figs. 5 and 6 in the main paper. I highly recommend moving Fig. 6 to the supplementary material, as it does not seem directly related to the main topic of this paper. If this paper were about a robust and powerful zero-shot generative model, then showing out-of-distribution cases (Fig. 6) of other methods would be reasonable. However, please note that the focus of this paper is on addressing the challenging short-focal length cases.
> >
> > Q2 and Q4 refer to a similar issue. I noticed that you did not include another important baseline in Table I of the rebuttal: w/o EFReg and w/o various focal length data in training. The reason I mentioned NFI and VQA-Diff is that I believe these studies have shown that adding various focal-length data during training can improve performance when dealing with short-focal-length scenarios. As they are prior works, I think it is necessary to demonstrate that Era3D processed one more step upon this. By doing so, we can see the necessity of EFReg more explicitly. The author should add this row to Table I and include the discussion in the main paper.
> >
> > [Weakness]
> >
> > I still have an issue with the response to Q5. Consider the classic MobileNet. When it was first proposed, do you think readers would have accepted it if the authors had only demonstrated that MobileNet requires fewer FLOPs and has a shorter inference time? Demonstrating performance is always important because there is usually a trade-off between performance and efficiency (though it is definitely desirable to improve both simultaneously).
> >
> > I understand that the resolution issue prevents the authors from comparing the methods using the current setup. However, I believe that for this type of comparison, it is acceptable to adjust the setup as long as the comparison remains fair. Specifically, the authors could train the models with a smaller resolution (perhaps on a smaller dataset as well, if it doesn’t compromise the results). In summary, fairness is the only consideration in this test, and for a top-tier conference, a thorough experiment is necessary.
> >
> > I will increase my rating if the authors can address the weakness mentioned in this comment. For now, I will keep it at 6.

---

> > > ### Comment · Area_Chair_qmKC · 2024-08-10
> > >
> > > Thank you for the discussion. Any other thoughts from other reviewers?

---

> > > ### Author Response · Authors · 2024-08-12
> > >
> > > Thanks for your valuable comments and suggestions.
> > >
> > > Regarding the suggestion of 'w/o EFReg and w/o various focal length data in training', we believe Wonder3D provides a good baseline in the main paper. They neither consider perspective distortion nor employ any specific dataset strategy.
> > >
> > > Following your comment on Q5, we recognize the importance of conducting performance comparisons between dense, epipolar, and our row-wise attention. Considering that in our orthogonal setup with the same elevation, epipolar attention is equivalent to row-wise attention (except for implementation differences), we compared dense attention and row-wise attention at a resolution of 256. We used the full 80,000 objects mentioned in the main paper for training. The training was conducted using 8 NVIDIA H800 GPUs, with a batch size of 128 for a total of 30,000 iterations. Each experiment took approximately 22 hours.
> > >
> > > The results reported in Table 3 demonstrate that row-wise attention can achieve comparable performance to dense attention. For the Chamfer distance, LPIPS, and PSNR metrics, our row-wise setting even outperforms the baseline. We attribute this to row-wise attention reducing the number of attention tokens, allowing the model to focus more on valuable tokens.
> > >
> > > **Table 3**: Performance of dense and row-wise attention at a resolution of 256.
> > > | Method   | CD ↓    | IoU ↑  | LPIPS ↓  | PSNR ↑ | SSIM ↑ |
> > > |----------|--------|--------|-------|-------|-------|
> > > | dense    | 0.0239 | **0.5877** | 0.140 | 20.73 | **0.819** |
> > > | rowwise  | **0.0232** | 0.5831 | **0.137** | **20.92** | 0.813 |
> > >
> > > We hope these findings address your concerns. If you have any further thoughts, we welcome active discussion and are committed to refining our paper accordingly.
> > >
> > > Thanks for your time and positive reviews for our work.

---

> > > > ### Comment · Reviewer_aLmz · 2024-08-12
> > > > **response to rebuttal 2**
> > > >
> > > > I have increased my rating to 7 as the authors have addressed most of my major concerns. I recommend that the authors reorganize Table 3 in the main paper by incorporating the content from Table 3 in the rebuttal.
> > > >
> > > > However, regarding the first point about the baseline, unless Wonder3D is exactly the same as this work, except for not considering perspective distortion and specific training data, the authors should refrain from making such a declaration. I still recommend that the authors add the corresponding row to Table 2 so that readers can find all relevant information in a single table.
> > > >
> > > > Good luck!

---

> > > > > ### Author Response · Authors · 2024-08-12
> > > > >
> > > > > We sincerely appreciate your great efforts in reviewing this paper. Your constructive advice and valuable comments really help improve our paper.  We will also reorganize the tables as suggested.
> > > > >
> > > > > Once more, we are appreciated for the time and effort you've dedicated to our paper!!!!

---

### Official Review · Reviewer_a9Xx · 2024-07-17

**Soundness:** 3
**Presentation:** 3
**Contribution:** 2
**Rating:** 5
**Confidence:** 4

**Summary:**

The authors proposed a method that can estimate the camera intrinsic matrix of the render of a given object, which attempted to solve the problem of other image-to-3d methods that only trained on fixed camera intrinsic matrix.

**Strengths:**

The proposed method is the first work that considers the change of the camera intrinsic matrixes for the image-to-3D generation, and proposed an attention strategy to reduce the computational cost.

**Weaknesses:**

1. I do not think row-wise attention should be a contribution as it is a special case of the epipolar attention from another paper.
2. Lacks the comparison with SV3D which is released before the deadline of the submission and can generate high resolution videos. Although Unique3D is released after the deadline, but it would be better if authors can also optionally include the comparison with it.
3. There is only a demo video in the supplementary material. If authors want refer reviewers to the arxiv version, it actually violates the double-blind policy.

**Questions:**

1. Authors mentioned that they used the feature maps of middle-level transformer block for intrinsic estimation. Can you specify it? Which level did you use? How many levels did you use? Any experiments about the impact of different levels?
2. Is it possible to directly generate images under perspective cameras?

**Limitations:**

1. I am not sure the novelty of the paper is enough for publication. The only contribution that I can tell is the camera intrinsics estimation. Row-wise attention is borrowed from other works. The authors mentioned that the proposed work can generate high resolution image, but sv3d can also do this.

---

> ### Author Rebuttal · Authors · 2024-08-06
>
> Thank you for your valuable time and insightful comments! We have tried to address your concerns in the updated manuscript and our rebuttal text:
>
> **Q1: Row-wise attention is just epipolar attention and is not the contribution of this paper.**
> We respectfully disagree with this characterization. As discussed in Line290 of the submission, the vanilla Eipolar attention adopted by previous methods significantly increases memory and time consumption due to their need for sampling points along the epipolar lines, which makes them even slower than the dense multiview attention. Era3D is the first attempt to simplify epipolar attention, which makes it extremely efficient for generating high-quality 3D objects. Thus, we consider it a core contribution of our work.
>
> **Q2: Lack of comparison with SV3D and Unique3D.**
> While we acknowledge the importance of providing comprehensive comparisons, Unique3D and SV3D are unpublished technical reports on arXiv. Both of them are concurrent works that released their codes near or after our submission. Despite it, we provide an additional comparison with Unique3D in Fig.2 of global response, which shows that Unique3D still suffers from perspective distortion while our method greatly alleviates this problem. We appreciate the reviewer's feedback regarding these comparisons.
>
> **Q3: There are no other supplementary materials other than the video and response to the reviewer's comment regarding the violation of the double-blind policy.**
> We respectfully disagree with the assessment. We strictly follow the NeurIPS 2024 guideline to attach the supplementary materials at the end of the main paper, including additional experiments and descriptions of our method. We also included an anonymous link at the end of the abstract to show our results without compromising the double-blind review process. We did not encourage or invite reviewers to search for our submission, and thus, we maintain that we have adhered to the double-blind reviewing policy as stipulated by the conference guidelines. We appreciate the reviewer's feedback and clarification on this matter.
>
> **Q4: Selection of features for EFReg.**
> We apologize for any lack of clarity in our presentation. For EFReg prediction, we utilize the final layer feature map of the intermediate UNet block with the lowest resolution since it could provide high-level global information.
>
> **Q5: Can the proposed method directly generate perspective images?**
> Our method is designed for 3D generation, which generates images around an object with orthogonal cameras. Thus, generating perspective images falls outside the scope of our paper. Moreover, generating perspective images will introduce additional challenges, such as varying scale prediction across multiple views, which would significantly increase the complexity of 3D generation. That's why existing works, including ours, focus on orthogonal setups rather than perspective ones.
>
> **Q6: Not sure the novelty of the paper is enough for publication.**
> As elaborated in Q1, our row-wise attention is distinct from the custom epipolar attention in previous studies. The proposed row-wise attention demonstrates a significant enhancement in training efficiency. SV3D is our concurrent work and requires considerably more resources for training. Therefore, we restate our contributions as follows:
> 1. **Era3D** is the first method that tries to solve the distortion artifacts in general 3D generation tasks;
> 2. We design the novel **EFReg** to enable diffusion models to take distorted images as inputs while outputting the orthogonal images on the canonical camera setting;
> 3. We propose row-wise multiview attention, an efficient attention layer for high-resolution multiview image generation;
>
> We believe our designs could advance the field and inspire other 3D generation models.

---

> > ### Comment · Reviewer_a9Xx · 2024-08-12
> > **Response to the rebuttal**
> >
> > I appreciate the feedback from the authors. I still have some concerns as follows:
> > 1. The author mentioned that "We leave the proof in the supplementary material.", but no such a thing.
> > 2. How come the final layer of the UNet has the lowest resolution, shouldn't it be the highest resolution?
> > 3. Can you please list the related methods that focus on orthogonal setups?

---

> ### Author Response · Authors · 2024-08-12
>
> Thank you for your valuable feedback! We try to address your issues in the following discussion.
>
> **Q1: Proof of Row-wise.**
> Please refer to the 'A.2 Proof of Proposition 1' section (on Page 18) in our initial submission for the detailed proof.
>
> **Q2: Selection of features for EFReg.**
> As mentioned in response to Q4, we select 'the final layer feature map of **the intermediate UNet block**' rather than 'the final layer feature map of the whole UNet'. Therefore, the feature has the lowest resolution.
>
> **Q3: Works on orthogonal setups.**
> The works on orthogonal setups include but are not limited to:
> + (**ICLR'2024**) MVDream: Multi-view Diffusion for 3D Generation.
> + (**Arixv'2024**) Unique3D: Unique3D: High-Quality and Efficient 3D Mesh Generation from a Single Image.
> + (**ECCV'2024**) CRM: Single Image to 3D Textured Mesh with Convolutional Reconstruction Model.
> + (**CVPR'2024**) EfficientDreamer: High-Fidelity and Robust 3D Creation via Orthogonal-view Diffusion Priors.
> + (**CVPR'2024**) Wonder3d: Single image to 3d using cross-domain diffusion.
>
> We can further address unclear explanations and remaining concerns if any.
>
> Once more, we appreciate the time and effort you've dedicated to our paper.

---

> > ### Comment · Reviewer_a9Xx · 2024-08-12
> >
> > I have adjusted my rate after further consideration.

---

> > > ### Author Response · Authors · 2024-08-13
> > >
> > > Thanks for your great efforts in reviewing our paper!!

---

### Official Review · Reviewer_5Kpg · 2024-07-23

**Soundness:** 3
**Presentation:** 2
**Contribution:** 3
**Rating:** 6
**Confidence:** 5

**Summary:**

This work introduces a novel take on multiview diffusion models, highlighting the potential to realize high-resolution images from one image. The method comes with a new design for the diffusion-based camera prediction module, focal length, and elevation of the input image elevation, together with row-wise attention in enforcing epipolar priors in the MV diffusion. The results show that the approach is very high in quality, detailed in 3D meshing ability, and multiview image generation with larger resolution ability while consuming much less computation compared to other presented approaches.

**Strengths:**

**Clear Motivation and Innovative Module Design**
In this paper, the authors address several significant challenges associated with MV Diffusion in 3D content generation. These challenges include issues such as low resolution, inefficient generation processes, and inconsistent camera settings. For each of these problems, the authors propose novel designs aimed at providing effective solutions.

**State-of-the-Art Results**
The authors claim to have achieved state-of-the-art performance in single-view image generation tasks, as evidenced by the results in Table 1 and Table 2. However, due to the rapid advancements in 3D generation, their work does not compare their generation quality with recent methods such as LRM.

**Novel Contribution to 3D MV Diffusion Generation**
To the best of my knowledge, this paper is the first to address and propose solutions for the distortion problem specifically in 3D MV Diffusion generation.

**Weaknesses:**

### Missing Critical Details
The paper lacks several essential points, including optimization time for inference, details on hyperparameters, and robustness testing.

### Insufficient Ablation Study
The ablation study does not adequately justify the architectural design choices. More comprehensive experiments are needed to support these decisions.

**Questions:**

1. The abstract (line 16) states, "Era3D generates high-quality... up to a 512×512 resolution while reducing computation complexity by **12x times**." I was really confused about this context. In the paper, the authors claim the design of row-wise multiview attention results in a 12-fold reduction in running time compared with Dense MV Attention. However, the author didn't show the complexity of other parts' time in their pipeline. Further Claims are needed.
2. Why did you choose viewpoints on an elevation of 0? Will it affect the generation for a specific side (e.g., bottom side)?
3. Have the authors tried to perform any sparse view reconstruction? And why you choose 6 views: {β, β + 45◦, β + 90◦, β − 45◦, β − 90◦, β + 180◦}? More ablation studies or claims may be needed.
4. I checked your demo video, but it only shows your result. Could you also compare it with other results in the video for better display?

**Limitations:**

Yes

---

> ### Author Rebuttal · Authors · 2024-08-06
>
> Thank you for your valuable time and insightful comments! We have tried to address your concerns in the updated manuscript and our rebuttal text:
>
> **Q1: Comparison with LRM-based methods.**
> Tab.1 showcases the comparisons of Era3D with OpenLRM and CRM on the GSO dataset, which shows that Era3D exhibits better performance than these two methods. Our method can be incorporated with LRM-based methods because recent LRM-based methods rely on multiview generation as the input while Era3D can provide generated multiview images.
>
> **Table 1**: Quantitative comparison with LRM-based methods:
>
> | Method | CD↓ | IoU↑ | LPIPS↓ | SSMI↑ | PSNR↑ |
> |--------|-----|------|--------|-------|-------|
> | openLRM | 0.0302 | 0.5243 | 0.158 | 0.738 | 19.03 |
> | CRM | 0.0237 | 0.5693 | 0.146 | 0.803 | 20.78 |
> | Ours | 0.0217 | 0.5973 | 0.126 | 0.873 | 22.74 |
>
> **Q2: Missing details of inference time and hyperparameters.**
> The whole process requires approximately 4 minutes, comprising 13 seconds for the multiview diffusion, 3 minutes for the NeuS reconstruction, and 10 seconds for the texture refinement. Our diffusion model employs a 6-layer MLP for pose estimation. We use the same hyperparameters as SD2.1, except what is explicitly stated in the 'Implementation Details' section. For the NeuS reconstruction, we use the same settings as Wonder3D. During texture refinement, we optimize the appearance for 200 iterations with a resolution of 1024 and a learning rate of 1e-3. We will add these details to the revision.
>
> **Q3: Absence of robustness testing.**
> We comprehensively evaluate the robustness of our pose prediction on the GSO dataset and the generation quality for in-the-wild objects in Tab.5, Tab.6, and Fig.11 of ' Supplementary Material'. We will also release our code for public testing and evaluation.
>
> **Q4: Why select generation viewpoints of elevation 0? Would this affect the reconstruction of the bottom?**
> In contrast to the setup with various elevations employed in Zero123++, Era3D is trained to generate six views at an elevation of 0. This choice is because row-wise attention requires a fixed elevation. Intuitively, models perceive an elevation of 0 more readily than a random elevation. While this setting may affect the objects with planar bottoms, such cases are uncommon. Note that the prior works such as SyncDreamer and Wonder3D also use similar pre-defined viewpoints.
>
> **Q5: Why only generate 6 views?**
> Generating dense views necessitates substantial memory for training. Fig.1 in global response illustrates sparse-view reconstruction results using 2, 4, and 6 views. Our method, utilizing 6 views, consistently produces complete and plausible 3D shapes.
>
> **Q6: Insufficient ablation of EFReg.**
> We utilize the intermediate feature map with the lowest resolution in UNet for EFReg prediction since it could provide high-level global information. The effectiveness of EFReg is evaluated qualitatively in Fig.7 of the main paper. To provide quantitative support, we additionally report the Chamfer Distance in Tab.2. These results confirm that EFReg facilitates robust pose estimation and enhances reconstruction accuracy.
>
> **Table 2**: Ablation of EFReg on GSO datasets with various elevation (α) and focal lengths (f). We report the Chamfer Distance (↓).
>
> | Pose | α=0 | | | | | f=∞ | | | | | α=0, f=∞ |
> |------|-----|-----|-----|-----|-----|-----|-----|-----|-----|-----|------------|
> |      | f=35 | f=50 | f=85 | f=105 | f=135 | α=-10 | α=10 | α=20 | α=30 | α=40 | |
> | w/o EFReg | 0.0237 | 0.0233 | 0.020 | 0.022 | 0.0217 | 0.0221 | 0.0217 | 0.0225 | 0.0231 | 0.0228 | 0.0217 |
> | w EFReg | 0.0223 | 0.0219 | 0.0216 | 0.0214 | 0.0214 | 0.0217 | 0.0216 | 0.0216 | 0.0219 | 0.0217 | 0.0213 |
>
> **Q7: Provide additional visual comparison with baselines in the demo video.**
> We appreciate the reviewer's suggestion. We will incorporate comparisons in the demo video to provide a more comprehensive visual presentation of our method's performance relative to existing approaches.
>
> **Q8: Confusing claim about computation complexity reduction of row-wise attention.**
> Compared to Dense MV attention, our row-wise attention reduces the computation complexity by 12 times. We will clarify this in the Abstract. In Tab.3, we list the memory usage and running time of each part of our pipeline, in which other parts include self attn, cross attn, and feed-forward layers. Notably, Dense MV attention layers constitute approximately 60% of the memory footprint and 75% of the running time in the overall pipeline. Our row-wise attention substantially mitigates these computational demands, with particularly remarkable improvements in execution time.
>
> **Table 3**: Memory usage and running time of the pipeline with 512 resolution and xFormer.
> | | Memory usage (G) | | | Running time (ms) | | |
> |----------|----------|----------|----------|----------|----------|----------|
> | | MV Attn | Other parts | Total | MV Attn | Other parts | Total |
> | Dense | 1.42 | ~1.0 | 2.40 | 22.96 | ~6.5 | 29.13 |
> | Epipolar | 1.71 | ~1.0 | 2.81 | 20.03 | ~6.5 | 26.75 |
> | Ours | 1.08 | ~1.0 | 2.09 | 1.86 | ~6.5 | 8.31 |

---

> ### Comment · Reviewer_5Kpg · 2024-08-13
> **Response to the rebuttal**
>
> I greatly appreciate the author’s detailed rebuttal, which effectively addressed nearly all of my concerns. However, based on my understanding of the field and feedback from another reviewer, I have some reservations about the paper’s novelty. Given the substantial body of existing literature in this area, I remain cautious about the contribution of this paper. Therefore, I will maintain my current score of 6.

---

> > ### Author Response · Authors · 2024-08-14
> >
> > We sincerely appreciate your great efforts in reviewing this paper. Your constructive advice and valuable comments really help improve our paper. We will add corresponding discussions in the revision.
> >
> > Once more, we are appreciated for the time and effort you've dedicated to our paper.

---

### Author Rebuttal · Authors · 2024-08-06

We thank the reviewers for their valuable comments. In summary, the reviewers are positive about the novelty, performance, and potential of our method: **"address several significant challenges associated with MV Diffusion"**(R-5Kpg), **"the first work that considers the change of the camera intrinsic matrixes for the image-to-3D generation"**(R-a9Xx) and **"be an interesting and novel solution"**(R-aLmz).

We include the necessary Figures and Tables in the attached PDF file. We respectfully direct the reviewers to the corresponding section for detailed response.

---

### Comment · Area_Chair_qmKC · 2024-08-09

Dear reviewers, do the authors' responses answer your questions or address your concerns? Thanks.

---

> ### Comment · Area_Chair_qmKC · 2024-08-12
>
> Dear reviewers, as we approach the final two days, please take a moment to review the author's responses and join the discussion. Thank you!

---

> > ### Comment · Area_Chair_qmKC · 2024-08-14
> >
> > Dear reviewers, the authors are eagerly awaiting your response. The author-reviewer discussion closes on Aug 13 at 11:59 pm AoE. Thanks!

---

### Decision · Program_Chairs · 2024-09-25

**Decision:**

Accept (poster)

**Comment:**

The paper introduces Era3D, a method for generating high-resolution multiview images from a single image using efficient row-wise attention and a diffusion-based camera prediction module. The method addresses key challenges in multiview diffusion, such as camera prior mismatch and computational inefficiency, and achieves state-of-the-art results in generating detailed 3D meshes.
We are glad to accept the paper because of its contribution to multiview image generation and all positive comments from reviewers.
The authors are still encouraged to include all concerns and suggestion from reviewers into their camera-ready version.